# Trajectories of Quality of Life among an International Sample of Women during the First Year after the Diagnosis of Early Breast Cancer: A Latent Growth Curve Analysis

**DOI:** 10.3390/cancers15071961

**Published:** 2023-03-24

**Authors:** Ruth Pat-Horenczyk, Lauren Kelada, Eleni Kolokotroni, Georgios Stamatakos, Rawan Dahabre, Gabriella Bentley, Shlomit Perry, Evangelos C. Karademas, Panagiotis Simos, Paula Poikonen-Saksela, Ketti Mazzocco, Berta Sousa, Albino J. Oliveira-Maia, Ilan Roziner

**Affiliations:** 1School of Social Work and Social Welfare, Hebrew University of Jerusalem, Jerusalem 9190500, Israel; lauren.kelada@mail.huji.ac.il (L.K.); rawan.dahabre@mail.huji.ac.il (R.D.); gabriell.bentley@mail.huji.ac.il (G.B.); perryshlomit@gmail.com (S.P.); 2Institute of Communication and Computer Systems, School of Electrical and Computer Engineering, National Technical University of Athens, 10682 Athens, Greece; ekolok@mail.ntua.gr (E.K.); gestam@mail.ntua.gr (G.S.); 3Department of Psychology, University of Crete, Foundation for Research and Technology, 70013 Heraklion, Greece; karademas@uoc.gr; 4Medical School, University of Crete, Foundation for Research and Technology, 70013 Heraklion, Greece; simosp@uoc.gr; 5Helsinki University Hospital Comprehensive Cancer Center, University of Helsinki, 00100 Helsinki, Finland; paula.poikonen-saksela@hus.fi; 6Department of Oncology and Hemato-Oncology, University of Milan, 20139 Milan, Italy; ketti.mazzocco@ieo.it; 7Applied Research Division for Cognitive and Psychological Science, European Institute of Oncology IRCCS, 20139 Milan, Italy; 8Breast Unit, Champalimaud Clinical Centre, Champalimaud Foundation, 1400-038 Lisboa, Portugal; berta.sousa@fundacaochampalimaud.pt; 9Champalimaud Research and Clinical Centre, Champalimaud Foundation, 1400-038 Lisboa, Portugal; albino.maia@neuro.fchampalimaud.org; 10NOVA Medical School, NMS, Universidade Nova de Lisboa, 1099-085 Lisboa, Portugal; 11Department of Communication Disorders, Sackler Faculty of Medicine, Tel Aviv University, Tel Aviv 69978, Israel; ilanr@tauex.tau.ac.il

**Keywords:** breast cancer, quality of life, trajectories, latent growth analysis, BOUNCE

## Abstract

**Simple Summary:**

The current study aimed to examine the quality of life among women coping with breast cancer during the first 12 months post-diagnosis. We followed 699 women from four different countries as part of the BOUNCE Project in order to learn about the various factors that may influence their well-being. We assessed the women every three months with questionnaires asking them to report on psychological, biological, and functioning aspects of their life. The results showed that four groups of patients could be distinguished: The largest group (47% of the participants) showed an initial medium level of quality of life and tended to improve with time during the first year after breast cancer diagnosis. The second group comprised about a quarter of the women (26%), who showed stability in their medium quality of life. The third group (18%) showed an initially high level of quality of life and tended to improve with time. Last, the smallest group (9%) reported an initial low quality of life that tended to remain stable over the first year, with no improvement. Thus, most women experienced improvements in QoL during the first year post-diagnosis. However, approximately one-third of women experienced a consistently low quality of life, and they need early interventions.

**Abstract:**

The current study aimed to track the trajectory of quality of life (QoL) among subgroups of women with breast cancer in the first 12 months post-diagnosis. We also aimed to assess the number and portion of women classified into each distinct trajectory and the sociodemographic, clinical, and psychosocial factors associated with these trajectories. The international sample included 699 participants who were recruited soon after being diagnosed with breast cancer as part of the BOUNCE Project. QoL was assessed at baseline and after 3, 6, 9, and 12 months, and we used Latent Class Growth Analysis to identify trajectory subgroups. Sociodemographic, clinical, and psychosocial factors at baseline were used to predict latent class membership. Four distinct QoL trajectories were identified in the first 12 months after a breast cancer diagnosis: medium and stable (26% of participants); medium and improving (47%); high and improving (18%); and low and stable (9%). Thus, most women experienced improvements in QoL during the first year post-diagnosis. However, approximately one-third of women experienced consistently low-to-medium QoL. Cancer stage was the only variable which was related to the QoL trajectory in the multivariate analysis. Early interventions which specifically target women who are at risk of ongoing low QoL are needed.

## 1. Introduction

Approximately 2.3 million women are diagnosed with breast cancer each year, making it the most commonly diagnosed cancer in the world [1]. Improvements in treatments and early detection have resulted in a lower mortality rate for women with breast cancer, meaning that there are more survivors than ever [2]. Global five-year survival rates for women with stage I, stage II, and stage III breast cancer are 0.86, 0.69, and 0.51, respectively [3]. Despite these improvements, the toll of the diagnosis and treatments can negatively impact women’s quality of life (QoL) [4], including their physical, psychological, and social well-being [5,6,7,8,9]. 

Many factors are associated with QoL after receiving a breast cancer diagnosis, including level of social support, coping, socioeconomic status, age, depression, type of cancer treatment, tumor size, and stage of diagnosis [10,11,12,13,14]. Nevertheless, the literature assessing factors associated with QoL has often been inconsistent. For example, some studies have found treatment via adjuvant chemotherapy to be negatively related to QoL [10], while other studies have not found a relationship between chemotherapy and QoL [12,15]. Moreover, longitudinal research shows that, for some women, their QoL declines after diagnosis but begins to rebound during survivorship [16,17,18]. While it was once believed that the majority of women would follow this trajectory of QoL, recent research is emerging to show that a significant portion of women will either continue to have low QoL or their QoL will continue to decline, even several years after diagnosis [14,19,20]. 

One reason for the inconsistent findings may be that previous research has often assessed the group mean for QoL over time. However, since women with breast cancer are heterogeneous, using the group average to determine the overall QoL trajectory may be inappropriate. Thus, it is important for researchers to use group-based trajectory modeling to account for individual variability and determine subgroups of individuals who follow common, distinct trends in QoL post-diagnosis [13,21]. In other words, research needs to determine the QoL trajectories of distinct subgroups of women with breast cancer [20]. While this method is increasingly used, researchers have concluded that further trajectory research is required with large samples and across multiple sites to be able to produce generalizable findings [13,20]. Such trajectory research is necessary, as it may allow clinicians to identify women who are at risk of experiencing ongoing low QoL and who are in need of further support [17,20].

Changes in the QoL trajectory over time among women with breast cancer may be conceptualized by using Sprangers and Schwartz’s (1999) [22] model of response shift and QoL after illness. In this model, antecedent variables (e.g., sociodemographic and personality) inform the response to a “catalyst”, which, in this case, is breast cancer. The responses include mechanisms such as coping strategies and response shifts, which include changes within individuals in their own internal standards, values, and conceptualization, and, together, this process leads to perceptions of QoL. This model helps to show the interplay between static characteristics (e.g., sociodemographics and personality traits) and modifiable characteristics (e.g., coping and mental health) in individual’s changing perceptions of QoL over time after a breast cancer diagnosis. Similarly, Lazarus and Folkman’s (1984) Transactional Model of Stress [23] shows that, after a stressor, personality, sociodemographics, appraisals/cognitions, emotions, and coping all lead to psychological (e.g., depression and anxiety) and long-term outcomes (e.g., QoL). 

Supporting theories of stress and QoL after illness, previous trajectories research found that various sociodemographic, clinical, and psychosocial factors are related to the group membership of QoL trajectory. For example, studies have found that coping, optimism, meaning, and low depression and anxiety symptoms are associated with having consistently high QoL [13,17]. Alternatively, maladaptive coping, low optimism, and high depression symptoms have been found to be predictors of a declining QoL trajectory [14,20]. However, findings have largely been inconsistent, particularly regarding whether age, chemotherapy, and surgery are related to a worse QoL trajectory [13,14,20]. As such, further research is necessary to be able to determine important predictors of the QoL trajectory. This may also benefit from including a range of antecedents and mechanisms to be able to explore the most salient predictors of the QoL trajectory over time for women with breast cancer. 

The current study is based on an international multicenter and prospective study on resilience in breast cancer patients as part of the project “BOUNCE” (https://www.bounce-project.eu/ (accessed on 4 February 2023)). The focus of the current study was to assess the trajectories of QoL among women with breast cancer by mapping their QoL every three months, from diagnosis to 12 months post-diagnosis. Our research aims were (1) to identify distinct QoL trajectories over the 12 months following diagnosis (e.g., stable, improving, or declining trajectories); (2) to identify the proportion of women who can be classified to each of these trajectories; and (3) to explore the sociodemographic, clinical, and psychosocial factors associated with the distinct trajectories. 

## 2. Materials and Methods

The BOUNCE project aimed at “Predicting Effective Adaptation to Breast Cancer to Help Women to BOUNCE Back” (visit https://www.bounce-project.eu/ (accessed on 5 February 2023) for project description). The sample included women diagnosed with breast cancer from four countries: Finland (Helsinki University Hospital), Israel (Shaare Zedek and Rabin Medical Centers, coordinated by the Hebrew University of Jerusalem), Italy (European Institute of Oncology), and Portugal (Champalimaud Clinical Centre).

### 2.1. Participants

The recruitment of participants to the study was suggested during the first clinical consultation following diagnosis with breast cancer. The inclusion criteria were (a) age between 40 and 70; (b) confirmed and operable invasive breast cancer; (c) tumor stage I-III; (d) receiving surgery or systemic treatment for breast cancer; and (e) understanding and signing the informed consent [24]. The exclusion criteria were (a) refusal to consent; (b) presence of metastases or history of another malignant cancer within the last five years; (c) history of early onset mental or severe neurologic disorder or any serious diseases within the last 12 months and no major surgery within four previous weeks; and (d) pregnancy or breastfeeding at the time of recruitment. The women who met the criteria were approached by the project research assistants and completed the informed consent forms, as approved by the local ethical committees. The response rate ranged between 74 and 78% between the four treatment centers. 

### 2.2. Measures

A variety of psychosocial, medical, and functional variables were collected using an online platform. Psychosocial, sociodemographic, lifestyle, and clinical variables were measured every 3 months, starting from pre-surgery assessment (M0, baseline) to 12 months (M12) after surgery. For the complete study protocol, see Pettini et al. (2022) [21].

Outcome measure: Quality of life. The level of QoL was measured by a single item from the European Organization for Research and Treatment of Cancer Quality of Life Questionnaire (EORTC QLQ-C30): “How would you rate your overall quality of life during the past week?” [25]. Participants answered this question on a scale from 1, “very poor”, to 7, “excellent”, with higher scores indicating a higher QoL. The research shows that single-item assessments of global QoL display strong test–retest reliability, discriminant validity, and convergent validity and are recommended to reduce the burden on distressed patients [26,27,28]. Single-item measures of QoL (also known as “ultra-brief” measures) have also been found to have good reliability, validity, and clinical value, as well as research value for longer measures [26,27].

Baseline sociodemographic and lifestyle characteristics were the country of residence; age; education level (coded 1 = “primary school” through 6 = “Doctoral-level studies”); marital status (alone vs. cohabiting with partner); number of children; employment status (part- or full-time employed vs. non-employed); monthly income (monetary sums coded “1” through “10” after adjustment to the Gross Domestic Product (GDP) income level of each country); amount of exercise (in minutes per week); adherence to a diet (coded as “no diet”; “Mediterranean diet”; or “other diet”, e.g., low-calorie, carb-free, fermentable oligosaccharides, disaccharides, monosaccharides and polyols- free (FODMAP-free)); smoking status (coded as “never smoked”, “smoked in the past”, or “smoking at present”); and alcohol consumption (calculated as amount of standard drinks per week basing on participants reports of frequency and amounts of beer, wine, and spirits consumption). 

Baseline general and disease-specific medical characteristics included presence of chronic diseases (yes vs. no); Body Mass Index (BMI); menopausal status (pre- vs. peri- vs. post-menopausal); family history of breast cancer (yes vs. no); leukocytes count; neutrophils count; serum creatinine; serum bilirubin; cancer stage; cancer grade; tumor molecular profile (Luminal A, Luminal B, Triple Negative, HER2 Enriched); progesterone receptor positivity; estrogen receptor positivity; HER2 positivity; Ki67 levels (≥25); and treatment type recommended at baseline (mastectomy, chemotherapy, radiotherapy, endocrine therapy, and anti-HER2 therapy). 

Psychological predictor variables. Regarding personality factors, the Life Orientation Test—Revised [29] was used to assess dispositional optimism and pessimism (six items; Cronbach’s α = 0.62 and 0.72, respectively); the Sense of Coherence Scale [30] to assess sense of coherence (13 items, Cronbach’s α = 0.81); the Connor Davidson Resilience Scale to assess trait resilience [31] (10 items; Cronbach’s α = 0.90); and the Mindful Attention Awareness Scale [32] to assess dispositional mindfulness (15 items; Cronbach’s α = 0.86). Regarding the resources for coping with cancer diagnosis and treatment, the Perceived Ability to Cope with Trauma Scale [33] was used to assess the relevant general ability (20 items assessing trauma and forward focused coping, and flexibility in coping; Cronbach’s α = 0.90 and 0.91, respectively). Regarding breast-cancer-related perceptions, the total score of the brief version of the Cancer Behavior Inventory [34] (12 items; Cronbach’s α = 0.89) was used to assess the self-efficacy to cope with cancer. Finally, emotional state was assessed with the 20-item version of the Positive and Negative Affectivity Schedule [35] (20 items; Cronbach’s α = 0.72 and 0.83, respectively), and with the Hospital Anxiety and Depression Scale [36] (HADS; Cronbach’s α = 0.84 and 0.81 for depression and anxiety subscales, respectively), while emotion regulation was assessed with the overall positive and overall negative regulation scores from the Cognitive Emotion Regulation Questionnaire [37] (18 items; Cronbach’s α = 0.81 and 0.70, respectively).

### 2.3. Statistical Analyses

After computing descriptive statistics and intercorrelations of research variables, we estimated a series of unconditional Latent Class Growth Models (LCGMs), using Mplus version 8.6 software [38]. There were missing values in the data (the minimal covariance coverage in the variance–covariance matrix used in the analyses was 0.70), and the data deviated from normality. Therefore, we used the MLR estimator that allows for maximum likelihood estimation in the presence of missing values. We fitted models with one-to-six latent classes of change in QoL over time. The QoL items for each of the five waves served as manifest variables in these models. In each model, the intercept, the linear slope, and the quadratic slope of change across time were estimated for each class. Following the recommendation of Nylund, Asparouhov, and Muthén’s (2007) [39] for the method of choosing the number of classes, we considered, for a relatively low value of log-likelihood, a relatively high entropy index, the smallest value of Bayesian Information Criteria (BIC), and a significant bootstrapped likelihood ratio test (BLRT). In addition, we preferred models with a non-negligible proportion of cases in the smallest class, based on estimated posterior probabilities. Last, we added covariates to the chosen model. The final model included only the predictors that emerged as significant in, first, univariate and then in the multivariate analysis.

## 3. Results

The sample for the current study was comprised of 699 participants who provided QoL reports for at least one wave of data collection, of whom 427 (61.1%) provided data for all five waves, 118 (16.9%) for four waves, 96 (14.9%) for two or three waves, and 58 (8.3%) for only one wave, the majority of which was for M0. Compared with participants retained in all the measurement occurrences, patients that missed at least one wave were significantly (*p* < 0.01) less educated, with lower employment rates and lower income; with higher rates of pre-existing chronic illnesses; lower proportions undergoing chemotherapy, radiotherapy, or endocrine therapy; and higher levels of depression and negative affectivity at baseline. The analyses presented here are based on all available data. 

The distribution of baseline sociodemographic, lifestyle, and general and disease-specific medical characteristics are shown in Table 1 and Table 2. The majority of the international sample was composed of married women (74%), both highly educated and employed, and the age average was approximately 55 years. Most of the women (nearly 90%) were diagnosed with early stages of I and II and received hormonal treatment radiotherapy and chemotherapy. The list of psychological predictors appears in Table 3. 

Overall quality of life, the main study outcome measure, showed a decline between baseline (*M* = 5.29, *SD* = 1.29) and M3 (*M* = 5.14, *SD* = 1.32) (*t*(583) = 3.71; *p* < 0.001) and then increased back to levels comparable to those at baseline (M6: *M* = 5.48, *SD* = 1.16; M9: *M* = 5.48, *SD* = 1.19; M12: *M* = 5.52, *SD* = 1.23), with all changes relative to M3 being significant at *p* < 0.001 and changes from M0 being non-significant (*p* = 0.14 and *p* = 0.17 for M6 and M9, respectively) or marginally significant (*p* = 0.033 for M12).

To assess the existence of distinct groups of change trajectories behind the overall trend described above, a series of unconditional Latent Class Growth Models with one-to-six classes was tested. The characteristics of each model appear in Table 4. Each model was estimated with three parameters: the intercept, the linear slope, and the quadratic slope. The four-class solution was chosen based on the combination of significant BLRT and a substantial proportion of smallest class (9%). This solution was preferred over the three-class one as depicting a more meaningful, in our judgment, clinical picture. This solution was preferred over the five-class one, which had only 2% of participants in the smallest class. 

In Table 5, the parameters of the Growth Curve Model for each of the four classes are presented. Class 1 is characterized by a medium–low baseline level (intercept) of reported QoL and nonsignificant rates of change across time (slopes); therefore, this group can be labeled medium and stable. Class 2 is characterized by a medium–high baseline level and a significant rate of linear change; therefore, this group can be labeled medium and improving. Class 3 is characterized by a relatively high baseline level and a significant rate of linear change; therefore, this group can be labeled high and improving. Moreover, in this class, the negative value of the quadratic slope is also significant, meaning that the improvement reaches an asymptote at some point. Finally, Class 4 is characterized by a relatively low baseline level of reported QoL and a nonsignificant slope of change across time; therefore, this group can be labeled low and stable (the statistical no significance result of the seemingly steep slope stems from a relatively high standard error of the parameter). The estimated trajectories of change in the four classes are represented graphically in Figure 1.

Finally, we attempted to predict the membership of the patients’ latent-class QoL trajectory by using the baseline sociodemographic and lifestyle characteristics, general and disease-specific medical characteristics, and psychological measures. First, univariate analyses were performed for each of these predictors. Only a few of these variables were related significantly (*p* < 0.05) to class membership: the country of residence, with χ^2^(9) = 30.43, *p* < 0.001; adherence to a diet, with χ^2^(6) = 15.06, *p* = 0.020; amount of exercise, with *F*(3678) = 6.17, *p* < 0.001; cancer stage, with χ^2^(6) = 13.68, *p* = 0.033; recommendation for radiotherapy, with χ^2^(3) = 13.75, *p* = 0.003; and the negative emotion regulation score from the Cognitive Emotion Regulation Questionnaire Scale (CERQ), with *F*(3684) = 4.66, *p* = 0.003.

At the second stage of the analysis, these six variables were entered into a stepwise multinomial logistic regression. Only one variable survived in this analysis: cancer stage, with Wald χ^2^(2) = 7.94, *p* = 0.019, which stemmed from a significant (*p* = 0.005) difference between participants with stage I cancer versus others. These patients were slightly less represented in the medium and stable (22.5% vs. 28.1) and the medium and improving (46.3% vs. 53.7%) classes and were slightly more represented in the high and improving (18.0% vs. 11.5%) and the low and stable (9.0% vs. 7.1) class—as compared to patients with cancer at stages II and III.

## 4. Discussion

The current study aimed to track the trajectory of QoL among subgroups of women with breast cancer in the first 12 months post-diagnosis. We also assessed the number and portion of women in each subgroup and the sociodemographic, clinical, and psychosocial factors related to the distinct trajectories. Our findings have implications for targeted, early interventions to improve and mitigate potential declines in QoL among women with breast cancer.

We identified four distinct subgroups of QoL trajectories: medium (or medium–low) and stable (26% of the participants); medium (or medium-high) and improving (47%); high and improving (18%); and low and stable (9%). Previous trajectory studies have also identified four or more subgroups with similar trajectories [13,17], while other studies have found only two trajectories [19,20]. Seemingly, studies with larger sample sizes (approximately >600 participants), including the current study, are able to identify more distinct QoL trajectories than studies with small sample sizes (<150 participants). Thus, it appears that four (or more) subgroups may be the most indicative of the general population.

We found that the two groups of participants that experienced relatively high levels of QoL post-diagnosis (Classes 2 and 3, 65% of the sample) also exhibited improvement over the subsequent 12 months. The remaining 35% of patients who experienced relatively low levels of QoL at baseline (Classes 1 and 4) remained stable over the 12 months post-diagnosis. It is important to note, however, that no class of participants was identified with declining QoL. 

Our findings are in line with previous research with patients in the United States which found that the majority of women experienced improvements in QoL in the year following diagnosis [17,19], but they are not consistent with research from Korea, which found that the majority of women experienced a low and stable QoL [20]. Ethnic and cultural differences between participants across studies may play a role in the differential findings [20]. In particular, illness perceptions may differ between cultures, which may influence QoL [40], meaning that interventions to improve QoL should take the patient’s cultural and country contexts into consideration [41].

As part of our third aim, adherence to a diet, amount of exercise, cancer stage, radiotherapy, negative emotion regulation, and country of residence were univariably related to the QoL trajectories. However, in the multivariable analyses, cancer stage was the only variable which was significantly related to the QoL trajectory. Specifically, patients with stage I cancer were less likely to have a low and stable or medium and improving QoL trajectory and were more likely to have a high and improving or low and stable QoL trajectory than women with stage II or stage III cancer. Other research has also found that advanced cancer stage predicts poorer QoL, though this has been inconsistently reported in the literature [13,42]. Alternatively, for some women with stage I breast cancer, the early stage may mean that friends, family, and perhaps even healthcare professionals underestimate their level of need for support. Kroenke et al. (2013) found that social support predicts QoL in different ways depending on the stage of diagnosis [43]. For instance, tangible support was related to QoL for women with late-stage cancer, while low affectionate support predicted poorer QoL for women with early stage breast cancer [43]. Taken together, the type of social support may moderate the relationship between the QoL trajectory and stage of diagnosis. Nevertheless, our finding that cancer stage was the only variable to differentiate between the QoL trajectories in the multivariable analysis is somewhat surprising in the context of previous research. As such, our findings should be replicated in order to verify whether cancer stage is a reliable predictor of QoL trajectory, above other potential predictors. 

### 4.1. Implications and Future Directions

Our findings show that the impact of breast cancer on women’s QoL can differ dramatically between women. Early identification of the probable trajectory of each woman’s QoL can guide clinicians to triage patients for QoL interventions. Personalized interventions can start soon after diagnosis and should focus on the enhancement of QoL throughout treatment and recovery. For example, physical and/or psychosocial interventions such as yoga or meaning-centered psychotherapy may be able to help improve QoL in women with breast cancer [44,45]. Furthermore, we found that a portion of women with stage I breast cancer can experience low and stable QoL, which shows that assumptions cannot be made regarding women’s QoL “recovery” after early stage breast cancer. We are unable to determine in the current study why some women with stage 1 breast cancer have improving QoL while others have declining QoL, and this should be an avenue for future research. Nevertheless, our study shows that QoL should be screened soon after diagnosis, regardless of stage of diagnosis, so that we can better allocate resources and intervention efforts for women at-risk of ongoing low QoL. 

Our findings contribute to the current uncertainty in the literature regarding sociodemographic and clinical predictors of QoL trajectory; some studies have found that clinical characteristics are related to low QoL trajectories [17,19], while other studies have found that sociodemographic or clinical factors are not able to differentiate QoL trajectories [13]. Further research is needed to determine whether there are reliable predictors of the QoL trajectory for women with breast cancer. Many other factors, including cultural differences between studies and differences in the healthcare systems across study sites, may help to explain the large variability in previous QoL trajectory research and should also be taken into consideration for future research. 

Future research could also use trajectory clustering, which is based on the latent growth curve analysis, as the basis for the development of artificial intelligence (e.g., machine learning, ML) models for the classification of a new diagnosis into one of the QoL trajectories identified in this study. Such ML models, following adequate training, would accept as input pertinent individual data of the subject and predict her QoL trajectory.

### 4.2. Limitations

The current study benefited from having a large sample of women with breast cancer who were recruited from multiple sites. Nevertheless, the study also had limitations. First, while our study was strengthened by the inclusion of four international sites, each site was located in a European country. Given that cultural contexts may influence women’s experiences with breast cancer and QoL, our results may not be generalizable beyond European populations. Second, we did not collect data on the racial characteristics of the participants, thus limiting the study’s specificity and generalizability. Third, QoL is multidimensional, and other psychosocial and/or physical factors are likely to be linked to the different QoL trajectories. Future research should assess more comprehensive measures of quality of life in order to inform interventions geared toward enhancing the quality of life for women who are at risk. 

## 5. Conclusions

This study suggests that there are four distinct QoL trajectories in the first 12 months after a breast cancer diagnosis. The majority of women will experience improvements over the course of the first year. However, approximately one-third of women will experience consistently low-to-medium QoL. Early interventions are needed to specifically target women who are at risk of ongoing low QoL. These interventions may tap into the physical, psychological, and/or social determinants of QoL in order to help ensure that all women can experience improvements in their QoL after a breast cancer diagnosis.

## Figures and Tables

**Figure 1 cancers-15-01961-f001:**
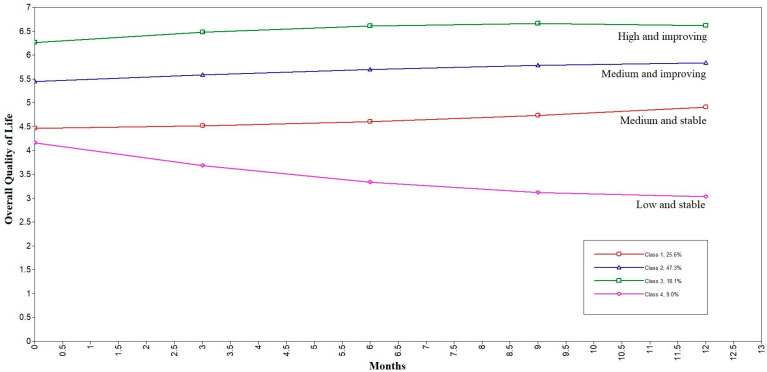
Trajectories of change in QoL in four classes (estimated means).

**Table 1 cancers-15-01961-t001:** Distribution of participant baseline sociodemographic and lifestyle characteristics (N = 699).

Characteristic	Values	N (%)	Mean (SD)
Country of data collection	Finland	225 (32.2%)	
	Israel	138 (19.7%)	
	Italy	189 (27.1%)	
	Portugal	147 (21.0%)	
Age	Years		54.92 (8.22)
Family status	Married/living with partner (vs. single, divorced, widowed)	517 (74.0%)	
Number of children	Range 0 to 12		1.95 (1.44)
Education level	Primary or secondary school	58 (8.3%)	
	High school or vocational diploma	235 (33.6%)	
	B.A.	237 (33.9%)	
	M.A. or higher	169 (24.2%)	
Employment	Full time, retired, self-employed (vs. unemployed, housewife, part-time)	563 (81.1%)	
Monthly Income (Euro)	1500 or lower	240 (36.5%)	
	1501–2500	286 (43.5%)	
	2501 or higher	131 (20.0%)	
Amount of exercise	Minutes per week		148.86 (157.67)
Adherence to diet	No diet	366 (52.8%)	
	Mediterranean	130 (18.8%)	
	Other	197 (28.4%)	
Smoking status	Never smoked	464 (66.9%)	
	Smoked in the past	131 (18.9%)	
	Current smoker	99 (14.2%)	
Alcohol consumption	Drinks per week		1.37 (2.41)
HADS Depression	Range 0–3		0.58 (0.51)
HADS Anxiety	Range 0–3		0.98 (0.58)

**Table 2 cancers-15-01961-t002:** Distribution of participant baseline general and disease-specific medical characteristics (N = 699).

Characteristic	Values	N (%)	Mean (SD)
Chronic diseases	Present (vs. absent)	233 (34.9%)	
BMI			25.72 (4.75)
Menopausal status	Pre- or peri- (vs. post-menopausal)	291 (42.2%)	
Family history of breast cancer	Present (vs. absent)	243 (34.8%)	
Leukocytes	10^3^/mcl		6.39 (1.92)
Neutrophils	10^3^/mcl		3.76 (1.65)
Creatinine	μmol/L		66.95 (10.22)
Bilirubin	μmol/L		9.06 (4.80)
Cancer stage	Stage I	333 (47.6%)	
	Stage II	289 (41.3%)	
	Stage III	77 (11.1%)	
Cancer grade	Grade I	130 (18.6%)	
	Grade II	362 (51.8%)	
	Grade III	207 (29.6%)	
Luminal A		529 (75.7%)	
Luminal B		77 (11.0%)	
Triple Negative		48 (6.9%)	
HER2 Enriched		26 (3.7%)	
Progesterone-receptor positivity		555 (79.4%)	
Estrogen-receptor positivity		623 (89.1%)	
HER2 positivity		121 (17.3%)	
Ki67 levels (≥25)		305 (43.6%)	
Type of treatment	Endocrine therapy	568 (81.3%)	
	Radiotherapy	523 (74.8%)	
	Anti HER2 therapy Chemotherapy Lumpectomy	108 (15.7%) 342 (49.5%) 497 (71.4%)	
	Mastectomy	193 (27.7%)	

**Table 3 cancers-15-01961-t003:** Distribution of psychological predictor variables (N = 699).

Characteristic	Scale Range	Mean (SD)
Dispositional optimism	0–4	2.78 (0.75)
Dispositional pessimism	0–4	1.34 (0.88)
Sense of Coherence Scale	1–7	5.17 (0.84)
Connor Davidson Resilience Scale	0–4	2.78 (0.70)
Mindful Attention Awareness Scale	1–6	4.39 (0.72)
Perceived Ability to Cope with Trauma Scale—trauma focus	1–7	5.15 (1.02)
Perceived Ability to Cope with Trauma Scale—forward focus	1–7	5.26 (0.86)
Cancer Behavior Inventory	1–9	7.15 (1.19)
Positive Affectivity Schedule	1–5	3.53 (0.73)
Negative Affectivity Schedule	1–5	2.00 (0.82)
HADS Depression	0–3	0.58 (0.51)
HADS Anxiety	0–3	0.98 (0.58)
Cognitive Emotion Regulation Questionnaire—positive	1–5	3.40 (0.70)
Cognitive Emotion Regulation Questionnaire—negative	1–5	2.14 (0.55)

**Table 4 cancers-15-01961-t004:** Comparison of Latent Class Growth Curve Models.

No. of Classes	Loglikelihood	BIC	*p* BLRT	Entropy	Proportion of the Smallest Class
1	−4724.75	9501.89			
2	−4385.88	8850.36	<0.001	0.74	0.30
3	−4278.25	8661.30	0.009	0.70	0.12
4	−4248.41	8627.81	0.021	0.65	0.09
5	−4211.61	8580.40	0.005	0.72	0.02
6	−4195.24	8484.96	0.449	0.72	0.02

**Table 5 cancers-15-01961-t005:** Parameters of Latent Growth Curve Models for the 4-class solution (means and standard errors). * *p* < 0.05, ** *p* < 0.01, *** *p* < 0.001

Class	Intercept	Linear Slope	Quadratic Slope	Proportion in Class
Class 1	4.47 *** (0.21)	0.01 (0.05)	0.002 (0.004)	0.256
Class 2	5.45 *** (0.09)	0.05 * (0.03)	−0.002 (0.002)	0.473
Class 3	6.26 *** (0.08)	0.09 *** (0.02)	−0.005 ** (0.002)	0.181
Class 4	4.16 *** (0.28)	−0.20 (0.11)	0.007 (0.008)	0.090

## Data Availability

The anonymized data that support the findings of this study are available upon request from the corresponding author. The data are not publicly available due to privacy and ethical restrictions.

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
