# Peer review of "Trajectories of Quality of Life among an International Sample of Women during the First Year after the Diagnosis of Early Breast Cancer: A Latent Growth Curve Analysis"

_cancers, 2023, doi:10.3390/cancers15071961_

Round 1

Reviewer 1 Report

This is an ambitious and ground-breaking study, and the authors are to be commended on the study design and the extensive variables collected on this longitudinal cohort.  My suggestions are ways to improve the presentation and make it easier for readers to follow, and also to enrich the discussion of the findings.

1) "and early detection have resulted in a lower mortality rate for women with breast cancer, meaning there are more survivors than ever [2]"   It would help readers get a general idea of survivorship, if survivorship rates were given for Stages 1-3, which corresponds to the sample in the study.  

2) "Many factors are associated with QoL after receiving a breast cancer diagnosis, including level of social support, coping, socioeconomic status, age, depression, and type of cancer treatment "  Given the physical and clinical status are mentioned later as strongly associated with QoL, it is strange that tumor size and stage are not included here (but are mentioned as one of the three pillars of Qol in the discussion).   As much as one might want to downplay the biology of cancer, it has a strong effect on QoL and of course, tumor recurrence and mortality.   These need to be stated as part of the context of Qol investigations.

3) It would help readers to have a definition of "survivorship", is it 1 year from diagnosis, 5 years?  How did your study define it?

4) Does the study really identify what differentiates the different classes one from another? “Goal 3) was to explore the sociodemographic, clinical and psychosocial factors associated with 109 the distinct trajectories".  Are distinct subgroups of women indeed identified so that a ML analysis would be able to identify them at cancer diagnosis besides the use of a single QoL question?  The results in fact reduce the findings to a single variable, which is cancer stage.  So, how does the study meet this goal?  This should be thoughtfully deliberated in the discussion. 

4)  I think you mean in exclusion criteria: "treatment for any OTHER major illness in the last half-year"

5)  It is very impressive that: variables were collected using an online platform (line 137).  How does this not exclude women who may not have access to a computer or who are not internet literate?  This jives with the relatively well educated sample.   Was any option given to help women with the platform or by having them fill out the variables while at the clinic in order to get help with the platform, which are often not user friendly.  I am assuming that they were set up in the language of the country where collected, but this is not mentioned.  The accessibility of the research platform/team is a key issue in research design.  I tried to access the study protocol from Pettini but it was not accessible at the website.

5a) It would help to know more details about the data collection procedures.  Who was in charge of recruitment?  MDs, nurses, research staff and who signed them on informed consent forms.

6) Psychological variables using scales.  Please give the range in the scale descriptions.  The results are presented in Table 3 as means and SDs with the readers having no sense what the range of the scale is. Please include them in Table 3.  You may prefer to include the Cronbach's alphas in the Table too to lighten up the text.

7)  "The sample for the current study was comprised of 699 participants who provided  QoL reports at least one wave of data collection, of whom 13% and 18% did not provide data at the 3- and 6-month follow-up assessments, respectively."  I do not understand how you could have included QoL data for respondents who did not have at least two data points out of the 4.  How can you calculate a trajectory from only one data point?   Unless you used a means-substitution treatment of missing data or some other method.  Please explain this conflict between the stated goals and your handling of the data where 13% and 18% were missing data.

8)  Indeed patients lost to follow-up constitute a unique group with lower rates of treatment, and with higher depression levels.  Perhaps it would be helpful to consider this group a fifth Class who get lost in the shuffle and are not available for follow-up.  This issue should also be noted in the discussion.

9)  Table 2 actually shows the proportion as well as the mean and should be labeled as such (column right). It would make it clearer if it were labeled Percentage (n) rather than N (%). 

10)  The results presented in the graph seem to differ strongly from the description in the text.  Two examples:

“Overall quality of life, the main study outcome measure, showed a decline between 2 baseline (M = 5.29, SD = 1.29) and M3 (M = 5.14, SD = 1.32), t(583) = 3.71, p < . 001, and then creased back to levels comparable to those at baseline. “   The only decline is apparent in Class 4 and the other three groups all increase to M3.

As mentioned in the text: Finally, Class 4 is characterized by a  relatively low baseline level of reported QoL and a nonsignificant slope of change across 2time and therefore, this group can be labeled low and stable (the statistical no significance  result of the seemingly steep slope stems from a relatively high standard error of the pa-rameter).  

As you note, the downward slope is so strong that the QoL decreases a full point on the scale.

Something just doesn’t add up.  Perhaps redraw the figure or check the results.  Or consider deleting the figure.  

Again, these results don’t seem to be reflected in how the figure looks and that it troubling.

My recommendation is acceptance after minor revisions as suggested in the points above.

Author Response

March 14, 2023

Dear Ms. Treena Guo
Managing Editor/Cancers
Email: treena.guo@mdpi.com

Dear Editor, 

We appreciate the positive and helpful comments made by the reviewers regarding our manuscript entitled “Trajectories of quality of life among an international sample of women during the first year after the diagnosis of early breast cancer: A Latent Growth Curve Analysis

Below, we provide a point-by-point explanation of our responses to each comment made by each of the reviewers. The corrections are highlighted in yellow within the manuscript.

Comments from Reviewer 1 and responses:

This is an ambitious and ground-breaking study, and the authors are to be commended on the study design and the extensive variables collected on this longitudinal cohort.  My suggestions are ways to improve the presentation and make it easier for readers to follow, and also to enrich the discussion of the findings.

Thank you!

Comment 1:  "and early detection have resulted in a lower mortality rate for women with breast cancer, meaning there are more survivors than ever [2]"   It would help readers get a general idea of survivorship, if survivorship rates were given for Stages 1-3, which corresponds to the sample in the study.  

Response: Thank you for this suggestion, we have now included the survival rates for stages 1-3:

“Global five-year survival rates for women with stage I, stage II and stage III breast cancer is, 0.86, 0.69 and 0.51, respectively [3].”

Comment 2:  "Many factors are associated with QoL after receiving a breast cancer diagnosis, including level of social support, coping, socioeconomic status, age, depression, and type of cancer treatment "Given the physical and clinical status are mentioned later as strongly associated with QoL, it is strange that tumor size and stage are not included here (but are mentioned as one of the three pillars of Qol in the discussion).   As much as one might want to downplay the biology of cancer, it has a strong effect on QoL and of course, tumor recurrence and mortality.   These need to be stated as part of the context of Qol investigations.

Response: We have now included tumor size and stage of diagnosis in this sentence:

“Many factors are associated with QoL after receiving a breast cancer diagnosis, including the tumor size, stage of diagnosis, type of cancer treatment, level of social support, coping, level of depression, socioeconomic status, and age [9-13].” (page

Comment 3: It would help readers to have a definition of "survivorship", is it 1 year from diagnosis, 5 years?  How did your study define it?

Response:

The current paper does not focus on survivorship and follows the women for 18 months from diagnosis of breast cancer.

Comment 4:  Does the study really identify what differentiates the different classes one from another? “Goal 3) was to explore the sociodemographic, clinical and psychosocial factors associated with 109 the distinct trajectories".  Are distinct subgroups of women indeed identified so that a ML analysis would be able to identify them at cancer diagnosis besides the use of a single QoL question?  The results in fact reduce the findings to a single variable, which is cancer stage.  So, how does the study meet this goal?  This should be thoughtfully deliberated in the discussion. 

Response: Thank you for this feedback. Our aim for this study was indeed to explore how sociodemographic, clinical and psychosocial factors were associated with the QoL trajectories, and we were somewhat surprised that only cancer stage remained significant in the multivariable analysis. We have now added more text in the discussion to deliberate how this goal was met, how our findings differ with some other previous research, and the directions for future research.

From the paragraph in the discussion summarizing our multivariable results: “Nevertheless, our finding that cancer stage was the only variable to differentiate between the QoL trajectories in the multivariable analysis is somewhat surprising in the context of previous research. As such our findings should be replicated in order to verify whether cancer stage is a reliable predictor of QoL trajectory, above other potential predictors.”

From the ‘implications and future directions’ section: “Furthermore, we found that a portion of women with stage I breast cancer can experience low and declining QoL, which shows that assumptions cannot be made regarding women’s’ QoL ‘recovery’ after early stage breast cancer. We are unable to determine in the current study why some women with stage 1 breast cancer have improving QoL while others have declining QoL, and this should be an avenue for future research. Nevertheless, our study shows that QoL should be screened soon after diagnosis, regardless of stage of diagnosis, so that we can better allocate resources and intervention efforts for women at-risk of ongoing low QoL.

Our findings contribute to the current uncertainty in the literature regarding socio-demographic and clinical predictors of QoL trajectory; some studies have found that clinical characteristics are related to low QoL trajectories [16, 18], while other studies have found that socio-demographic or clinical factors are not able to differentiate QoL trajectories[12]. Further research is needed to determine whether there are reliable predictors of QoL trajectory for women with breast cancer. Many other factors including cultural differences between studies and differences in the healthcare systems across study sites may help to explain the large variability in previous QoL trajectory research and should also be taken into consideration for future research. “

Comment 5: I think you mean in exclusion criteria: "treatment for any OTHER major illness in the last half-year"

Response: Thank you. This change has been made.

Comment 6: It is very impressive that: variables were collected using an online platform (line 137).  How does this not exclude women who may not have access to a computer or who are not internet literate?  This jives with the relatively well educated sample.   Was any option given to help women with the platform or by having them fill out the variables while at the clinic in order to get help with the platform, which are often not user friendly.  I am assuming that they were set up in the language of the country where collected, but this is not mentioned.  The accessibility of the research platform/team is a key issue in research design.  I tried to access the study protocol from Pettini but it was not accessible at the website.

Response: Thank you. We updated the reference of Pettini et al. (2022) which is published in OA.

Pettini G, Sanchini V, Pat-Horenczyk R, Sousa B, Masiero M, Marzorati C, Galimberti V, Munzone E, Mattson J, Vehmanen L, Utriainen M, Roziner I, Lemos R, Frasquilho D, Cardoso F, Oliveira-Maia A, Kolokotroni E, Stamatakos G, Leskelä R, Haavisto I, Salonen J, Richter R, Karademas E, Poikonen-Saksela P, Mazzocco K. (2022).Predicting Effective Adaptation to Breast Cancer to Help Women BOUNCE Back: Protocol for a Multicenter Clinical Pilot Study

JMIR Res Protoc 2022;11(10):e34564. URL: https://www.researchprotocols.org/2022/10/e34564. DOI: 10.2196/34564

Comment 7: It would help to know more details about the data collection procedures.  Who was in charge of recruitment?  MDs, nurses, research staff and who signed them on informed consent forms

Response:

We added to the method a short description of the recruitment method: “

Comment 8: Psychological variables using scales.  Please give the range in the scale descriptions.  The results are presented in Table 3 as means and SDs with the readers having no sense what the range of the scale is. Please include them in Table 3.  You may prefer to include the Cronbach's alphas in the Table too to lighten up the text.

Response:

The scale ranges are now added to Table 3

Comment 9: "The sample for the current study was comprised of 699 participants who provided QoL reports at least one wave of data collection, of whom 13% and 18% did not provide data at the 3- and 6-month follow-up assessments, respectively."  I do not understand how you could have included QoL data for respondents who did not have at least two data points out of the 4.  How can you calculate a trajectory from only one data point?   Unless you used a means-substitution treatment of missing data or some other method.  Please explain this conflict

Response:

Indeed, we used the MLR estimator that allows for maximum likelihood estimation in the presence of missing values, i.e., we used all of the available data in our analyses. We have added this point in an explicit manner in the Statistical Analyses section.

Comment 10: between the stated goals and your handling of the data where 13% and 18% were missing data

 Indeed, patients lost to follow-up constitute a unique group with lower rates of treatment, and with higher depression levels.  Perhaps it would be helpful to consider this group a fifth Class who get lost in the shuffle and are not available for follow-up.  This issue should also be noted in the discussion.

Response:

We were not clear in our description of missingness in our data. We are now more precise in describing missingness in the first paragraph of the Results section. As stated there and in our response to comment 10 above, we used data provided by all Pts in the analyses, thus the classes we describe are inclusive of all the sample.

Comment 11: Table 2 actually shows the proportion as well as the mean and should be labeled as such (column right). It would make it clearer if it were labeled Percentage (n) rather than N (%). 

Response: In response to feedback from Reviewer 2, we have now edited the tables so that portions and percentages are in a separate column to the means/SDs.

Comment 12: The results presented in the graph seem to differ strongly from the description in the text.  Two examples: “Overall quality of life, the main study outcome measure, showed a decline between 2 baseline (M = 5.29, SD = 1.29) and M3 (M = 5.14, SD = 1.32), t(583) = 3.71, p < . 001, and then creased back to levels comparable to those at baseline. “The only decline is apparent in Class 4 and the other three groups all increase to M3.

As mentioned in the text: Finally, Class 4 is characterized by a relatively low baseline level of reported QoL and a nonsignificant slope of change across 2time and therefore, this group can be labeled low and stable (the statistical no significance result of the seemingly steep slope stems from a relatively high standard error of the parameter).  

As you note, the downward slope is so strong that the QoL decreases a full point on the scale.

Something just doesn’t add up.  Perhaps redraw the figure or check the results.  Or consider deleting the figure.  

Again, these results don’t seem to be reflected in how the figure looks and that it troubling.

Response:

Thank you for this important comment. In response to this comment and a comment by Reviewer 1, we checked text for inconsistencies and made sure that the same labels are used in the Results and the Discussion sections, also changing the verbiage in such a way as to make the interpretation of the results easier. The discrepancy between what is seen in the Figure and the results of comparing the means at each wave stems from the fact that in the Figure, not actual, but rather estimated means are presented. The text and the figure caption were changed accordingly. It's the same type of seeming contradiction as the seemingly steep negative slope in class 4 which was found to be non-significant in the statistical analysis. We still believe that it's better to keep the Figure as it adds information in spite of the seeming contradiction unless the Reviewer or the Editor deems otherwise.  

Reviewer 2 Report

Dear authors, dear editor,

thank you for giving me the opportunity to review this interesting and well-written paper. I agree with the authors that beyond looking at group means of quality of life, considering the development of QoL over time, as well as the identification of subgroups according to their needs and resources can give us important further insights into the topic.

Here are some points that I would like to address:

1. In the Abstract and Highlights, the factor(s) associated with the latent classes (stage) might be mentioned, as this was also a goal of the study.

2. Measures: I am not quite sure whether I understood something wrong - In l. 139 it is stated that the study rounds were from pre-surgery to 18 months after surgery. In the results, only the course over 12 months (5 waves) is reported (month 0 = surgery?). Maybe you can add a statement on which 12 months you refer to and why pre-surgery data was not used (it would be very interesting to see which part of later QoL variance is explained by pre-existing problems and which part by treatments).

3. On p. 4 ll. 155 and 157 you might want to introduce the abbreviations "GDP" and "FOFMAP".

4. p. 4, l. 167: Did I understand correctly that it was not assessed whether patients really had the planned treatments? For example, treatments might have been cancelled early due to side effects or another treatment might have been chosen if the first one was not successful enough. In the results, e.g. p. 5 l. 215 it is phrased "received" treatments. To be more precise, this could be replaced by "planned" or "recommended" treatments.

5. Please clarify the beginning of the results, p. 5 l. 206: Participants "lost to follow up" were all those who missed at least one measurement? Were there only missings at 3 and 6 months, but not at 9 and 12 months?

6. The tables are hard to read, I would recommend to align the text columns to the left and the data columns to the right or at least equally. The space would even allow for 2 columns, one for continuous variables (mean/SD) and one for N/%.

7. The clusters and their naming are not consistent. Class 4 is named "low and stable" in the description (l. 255 and the previous text) but "low and declining" in l. 279. From the graph, I would also think that it is declining?

8. Figure 1: Add a label to the y-axis. Was the C30 score not transformed to 0-100 as recommended in the manual? And the x-axis might also get a label (months since surgery?) and end at 12? The legend with the percentages is hard to read. The difference between "medium and improving" and "medium and stable" is hard to detect. The lines are almost parallel. Although the red line ("medium and stable") starts at a lower level, I would think that it increases even more than the blue line ("medium and improving")?

9. I would find it interesting to include a table with the results on the predictors of the QoL trajectory (if the number of tables is limited, maybe rather include the current Table 4 only as supplementary material)

10. Discussion, p. 9 l. 310: As the psychological and sociodemographic variables did not "survive" the multivariate analysis, I think it is misleading to claim that these variables differentiated the four QoL classes. In fact, these relationships were obviously mediated by stage.

11. In the implications, you might focus more on differential needs to enhance QoL. Obviously, stage I patients get along well and recover without further specific intervention, whereas stage II and III might need more support? The fact that stage is related to QoL overall could also be a result of a "normal" multivariate analysis. What is the clinical implication of the specific latent class approach regarding the trajectory of QoL? Are there specific implications for patients with declining or increasing QoL?

Author Response

March 14, 2023

Dear Ms. Treena Guo
Managing Editor/Cancers
Email: treena.guo@mdpi.com

Dear Editor, 

We appreciate the positive and helpful comments made by the reviewers regarding our manuscript entitled “Trajectories of quality of life among an international sample of women during the first year after the diagnosis of early breast cancer: A Latent Growth Curve Analysis

Below, we provide a point-by-point explanation of our responses to each comment made by each of the reviewers. The corrections are highlighted in yellow within the revised manuscript.

Reviewer 2:                

Thank you for giving me the opportunity to review this interesting and well-written paper. I agree with the authors that beyond looking at group means of quality of life, considering the development of QoL over time, as well as the identification of subgroups according to their needs and resources can give us important further insights into the topic.

Comment 1: In the Abstract and Highlights, the factor(s) associated with the latent classes (stage) might be mentioned, as this was also a goal of the study.

Response: The following sentence has now been added to the abstract:

“Cancer stage was the only variable which was related to QoL trajectory in the multivariate analysis”.

The following point has also now been added to the highlights:

“Cancer stage is related to QoL trajectory.”

Comment 2: Measures: I am not quite sure whether I understood something wrong - In l. 139 it is stated that the study rounds were from pre-surgery to 18 months after surgery. In the results, only the course over 12 months (5 waves) is reported (month 0 = surgery?). Maybe you can add a statement on which 12 months you refer to and why pre-surgery data was not used (it would be very interesting to see which part of later QoL variance is explained by pre-existing problems and which part by treatments).

Response:

Mentioning of M18 was a typo. Thank you for noticing this. The text is amended now so that the correct label for the last wave of data collection (M12) is used throughout the text.

Comment 3: On p. 4 ll. 155 and 157 you might want to introduce the abbreviations "GDP" and "FOFMAP".

Response: Thank you, we have now made these changes:

“Gross Domestic Product (GDP) income level of each country); amount of exercise (in minutes per week); adherence to a diet (coded as "no diet", "Mediterranean diet" or "other diet", e.g., low-calorie, carb-free, fermentable oligosaccharides, disaccharides, monosaccharides and polyols- free (FODMAP-free)…”

Comment 4: p. 4, l. 167: Did I understand correctly that it was not assessed whether patients really had the planned treatments? For example, treatments might have been cancelled early due to side effects or another treatment might have been chosen if the first one was not successful enough. In the results, e.g. p. 5 l. 215 it is phrased "received" treatments. To be more precise, this could be replaced by "planned" or "recommended" treatments.

Response:

Thank you for your comment. We changed the phrasing to "recommended treatments" as suggested. 

Comment 5: Please clarify the beginning of the results, p. 5 l. 206: Participants "lost to follow up" were all those who missed at least one measurement? Were there only missings at 3 and 6 months, but not at 9 and 12 months?

Response:

Thank you for this important comment. We were imprecise in describing the missingness in our data. Indeed, the Pts labelled "lost to follow up" were the ones who didn't participate in all measurements. We now describe the missingness rates and characteristics of those not providing the full data more clearly.   

Comment 6: The tables are hard to read, I would recommend to align the text columns to the left and the data columns to the right or at least equally. The space would even allow for 2 columns, one for continuous variables (mean/SD) and one for N/%.

Response: We have now aligned the content of the tables to the left, and have created a separate column for the means/SDs.       

Comment 7: The clusters and their naming are not consistent. Class 4 is named "low and stable" in the description (l. 255 and the previous text) but "low and declining" in l. 279. From the graph, I would also think that it is declining?

Response:

We checked the text thoroughly and made sure that all the class labels are consistent and informative regarding the statistical findings. Regarding Class 4: We were aware of the discrepancy between the visual characteristics of the graph and the statistical results and therefore, the following appeared in the presentation of our results: "(the statistical no significance result of the seemingly steep slope stems from a relatively high standard error of the parameter)".

Comment 8: Figure 1: Add a label to the y-axis. Was the C30 score not transformed to 0-100 as recommended in the manual? And the x-axis might also get a label (months since surgery?) and end at 12? The legend with the percentages is hard to read. The difference between "medium and improving" and "medium and stable" is hard to detect. The lines are almost parallel. Although the red line ("medium and stable") starts at a lower level, I would think that it increases even more than the blue line ("medium and improving")?

Response:

We added the Y and X axes labels. As we used a single item from the C30 scale (item 30, the overall QoL assessment), thus there was no need to transform the score. We now mention this fact once more in the Methods section. We changed somewhat the figure format and hope it is more readable. Regarding the seeming improvement of the red line – parallel to our response to comment 7 above, the statistical significance of effects is not completely reflective of the trajectories' visual properties.

Comment 9: I would find it interesting to include a table with the results on the predictors of the QoL trajectory (if the number of tables is limited, maybe rather include the current Table 4 only as supplementary material)

Response:

Since only a single predictor of QoL trajectory was found to be statistically significant in the multivariate analysis, we consider it uninformative to add the multivariate results. Of course, if the Reviewer or the Editor regard this as necessary, we will add such a table. Note that as more than 40 potential predictors were tested, this table will be quite long.

Comment 10: Discussion, p. 9 l. 310: As the psychological and sociodemographic variables did not "survive" the multivariate analysis, I think it is misleading to claim that these variables differentiated the four QoL classes. In fact, these relationships were obviously mediated by stage.

Response: We have now reworded this sentence to de-emphasize the univariable findings:

As part of our third aim, adherence to a diet, amount of exercise, cancer stage, radiotherapy, negative emotion regulation, and country of residence were univariably related to the QoL trajectories.  However, in the multivariable analyses, cancer stage was the only variable which was significantly related to QoL trajectory.

Comment 11: In the implications, you might focus more on differential needs to enhance QoL. Obviously, stage I patients get along well and recover without further specific intervention, whereas stage II and III might need more support? The fact that stage is related to QoL overall could also be a result of a "normal" multivariate analysis. What is the clinical implication of the specific latent class approach regarding the trajectory of QoL? Are there specific implications for patients with declining or increasing QoL?

Response: Thank you for this thoughtful feedback. We have now added in a further discussion around the implications of our finding that cancer stage is related to QoL trajectory, including implications for future research:

“Furthermore, we found that a portion of women with stage I breast cancer can experience low and declining QoL, which shows that assumptions cannot be made regarding women’s’ QoL ‘recovery’ after early stage breast cancer. We are unable to determine in the current study why some women with stage 1 breast cancer have improving QoL while others have declining QoL, and this should be an avenue for future research. Nevertheless, our study shows that QoL should be screened soon after diagnosis, regardless of stage of diagnosis, so that we can better allocate resources and intervention efforts for women at-risk of ongoing low QoL.”

Round 2

Reviewer 2 Report

Dear authors,

thank you very much for the revised manuscript and the diligent consideration of all aspects. I just have some minor cosmetic suggestions left:

- l. 33: Either write the full word "Quality of life" or introduce the abbreviation after the first naming in l. 22

- l. 35 "low to medium quality of life"?

- l. 223: closing parenthesis missing

- Table 1 is much clearer now, but for education and income, something still went wrong with the formatting

- Thank you for explaining discrepancy between statistics and the apparent "curve" in the "low and stable" QoL class! However, the class is still called “low and declining” in l. 297, 324, 334, and 357 of the revised manuscript.

Author Response

Thank you very much for the additional corrections. 

- l. 33: Either write the full word "Quality of life" or introduce the abbreviation after the first naming in l. 22

Response: I avoided abbreviations of the terms in the simple summary as indicated. 

After the first spelling of Quality of Life, we used the abbreviation QLF in the following text.

- l. 35 "low to medium quality of life"?

Response: We deleted the "low to medium quality of life" and kept only "low quality of life" 

- l. 223: closing parenthesis missing. 

Response: DONE, thank you!

- Table 1 is much clearer now, but for education and income, something still went wrong with the formatting

Response: Corrected the format of the table

- Thank you for explaining the discrepancy between statistics and the apparent "curve" in the "low and stable" QoL class! However, the class is still called “low and declining” in l. 297, 324, 334, and 357 of the revised manuscript.

Response: We are now consistent in using a "low and stable" trajectory and not a "low and declining" trajectory.  I corrected the five times that "declining" had appeared. 
